# The Association Between Internet Addiction and Adolescents’ Mental Health: A Meta-Analytic Review

**DOI:** 10.3390/bs15020116

**Published:** 2025-01-23

**Authors:** Elena Soriano-Molina, Rosa M. Limiñana-Gras, Rosa M. Patró-Hernández, María Rubio-Aparicio

**Affiliations:** 1Department of Personality, Evaluation and Psychological Treatment, Faculty of Psychology and Speech Therapy, University of Murcia, 30100 Murcia, Spain; liminana@um.es (R.M.L.-G.); rosapatro@um.es (R.M.P.-H.); mrubioaparicio@um.es (M.R.-A.); 2School Psychology Department, European School of Alicante (EU), 03540 Alicante, Spain

**Keywords:** internet addiction, problematic internet use, adolescents, mental health, internalizing problems, externalizing problems

## Abstract

This study examines the association between problematic internet use, or internet addiction, and adolescent mental health, focusing on key psychological variables, assessing the strength of these associations, and identifying potential moderating factors. Methods: A search of the Web of Science databases over the past five years identified 830 articles. Of these, 33 met the inclusion criteria, involving 303,243 participants (average age 14.57; 49.44% female). The selection process was verified by two researchers. Results: Nine psychological variables were analyzed: depression, anxiety, stress, suicidal behaviour, psychological well-being, self-esteem, externalizing problems, aggressiveness, and impulsiveness. Internet addiction showed positive correlations with aggressiveness (*r*+ = 0.391), depression (*r*+ = 0.318), anxiety (*r*+ = 0.252), and suicidal behaviour (*r*+ = 0.264). Negative correlations were observed with psychological well-being (*r*+ = −0.312) and self-esteem (*r*+ = −0.306). No significant associations were found for externalizing problems, impulsiveness, or stress. None of the moderators showed a significant correlation with internet addiction and depression. Conclusions: Although limited by small sample sizes for some variables and the cross-sectional design of most studies, the findings confirm that there is a negative relationship between internet addiction and adolescent mental health. It is related to poorer self-perceived health, greater psychological distress, and greater aggression.

## 1. Introduction

The use of the internet has become a daily practice, present in almost all areas of our lives, from work or school activities to commerce, food, leisure, and entertainment, among many others. Therefore, it is currently difficult to find, consume, or carry out activities that are not linked to the internet. Since its inception in 1983, the number of users using this tool has grown exponentially to reach the latest figure, recorded at the end of 2023, of 5400 million users worldwide (around 67.4% of the total population) ([93]).

One of the reasons why the number of internet users has grown so much in such a short time are the great advantages and benefits the internet provides, such as access to updated information, a greater possibility of communicating with other people, ease and convenience when shopping, working remotely, etc. However, despite all these positive aspects, there are also many other negative aspects related to its use, due precisely to its accessibility characteristics and the wide range of services that this tool offers ([86]). This accessibility can lead to overuse, thus increasing the risk of the appearance of the so-called problematic internet use (PIU) or internet addiction (IA).

Both terms—internet addiction and problematic internet use—are defined as a use of the internet that generates psychological, academic, social, or occupational difficulties in a person’s life ([92]). Although not officially recognized in diagnostic systems such as the Diagnostic and Statistical Manual of Mental Disorders (DSM-V), these terms share characteristics of addictive behaviours, including an excessive use of the internet, withdrawal symptoms, tolerance, and adverse consequences at the interpersonal, intrapersonal, or daily activity level. Thus, these phenomena have an evident potential to generate significant psychological damage ([92]).

A key distinction in the study of PIU lies in differentiating between generalized PIU (GPIU) and specific PIU (SPIU). GPIU is characterized by a broad problematic pattern that is not linked to specific activities, where the individual engages in general inappropriate use of the internet without a specific purpose, using this network as a means to interact with the outside world due to difficulties in the social sphere ([28]; [55]; [68]). This review will focus primarily on this type of PIU.

On the other hand, SPIU is related to behaviours focused on specific activities, such as online games or online shopping, and is usually influenced by pre-existing clinical conditions that promote these behaviours, exacerbated by internet use. In these cases, if the individual could not express their problem through the internet, they would probably do so through another medium ([28]; [55]; [68]). In the case of online gaming, one of the most researched SPIUs includes terms such as Internet Gaming Disorder (IGD), which is considered a non-substance addictive disorder whose main characteristic is the recurrent and persistent participation in online video games over time, leading to clinically significant distress ([16]), or terms such as Problematic Online Gaming (POG), which allude to a problematic use of this practice ([50]). [68] ([68]) state that although there is some relation between PIU and IGD, they are distinct concepts, as IGD is a specific disorder that represents only one part of the PIU spectrum. Therefore, focusing exclusively on IGD could exclude other important aspects of PIU that also affect adolescents’ quality of life.

Another key aspect to consider is the impact of the COVID-19 pandemic. However, it should be noted that it represents a unique context that significantly disrupted usual patterns of social interaction and social media use, atypically influencing mental health issues. According to [78] ([78]), lockdown and social distancing measures increased reliance on digital platforms for work and social interaction, exacerbating stress and exposure to content that negatively affects self-image. This increase in psychological stress, along with anxiety and loneliness, contributed to addiction to the internet and platforms such as Instagram, resulting in an excessive and problematic use of social media as a coping mechanism during the pandemic ([6]).

In this sense, [103] ([103]) highlight that, during the pandemic, social media played a dual role: on the one hand, it served as a channel for obtaining social support in an environment marked by physical isolation, and on the other, its excessive use as a coping mechanism was associated with lower levels of emotional well-being. This duality reflects a complexity that differs from the dynamics observed in non-pandemic conditions.

Furthermore, previous meta-analytic studies ([38]; [103]) have found that the relationship between social media and well-being during the pandemic is not significant, in contrast to research conducted in pre-pandemic contexts ([70]; [107]), where consistent negative correlations were reported. This suggests that the pandemic context introduced specific contextual factors, such as increased loneliness and reliance on social media to maintain meaningful social interactions ([12]; [75]), which would attenuate the association between social media and mental health problems.

Electronic devices, such as mobile phones or tablets, are deeply integrated into everyday life and closely linked to internet use, making them susceptible to the development of problematic internet use ([80]). However, in this study, no distinction has been made between the different devices, as the focus has been on a generalized analysis of the problem, mainly addressing GPIU given its transversal nature and global impact on various areas of life ([55]). However, in the Discussion section, some specific results related to the use of mobile phones are considered to contextualize the findings obtained.

Therefore, the term IA will be used in this study to refer to internet addiction (IA) itself, to problematic internet use (PIU), and to the generalized use of any digital technology (GPIU), excluding online gaming, which represents a specific form of internet use categorized as an addictive disorder.

Adolescents are the most vulnerable group to developing this addiction, as, at this stage of life, they are particularly sensitive to the social environment in which they live. Easy access through devices such as mobile phones or tablets makes internet use an important part of their daily lives ([59]; [18]). Recent studies show an increase in internet addiction, particularly in regions such as East Asia, Europe, and North America, where internet use is more common. In some Asian countries, it is estimated that up to 47.4% of adolescents may be affected by this problem. On the other hand, in Western countries, rates range from 1.6% to 5.5%, depending on factors such as the diagnostic criteria used and cultural particularities ([73]).

Through this technology, adolescents have a means of learning, communication, leisure, and fun, making this technology an essential part of their daily lives. Thanks to the internet, communication with peers also becomes easier, since physical distance between interlocutors is no longer a problem. In addition, anonymity and the absence of eye contact facilitate interaction, masking identity and facilitating the communication of unpleasant topics or emotions ([18]; [85]). Likewise, some difficulties specific to adolescence have been pointed out, such as omnipotence, the tendency to externalize guilt, the search for sensations, a lack of self-control, the difficulty of recognizing addictive behaviours, or the normalization of risky behaviours. These, together with adolescents’ limited life experience, can increase their risk of developing addictions ([18]; [85]).

The core issue of problematic internet use, or internet addiction, lies in the future repercussions that this misuse has on the mental health of these adolescents ([56]), hence the importance of education for the development of good habits during this stage ([24]).

In general, adolescence is a particularly sensitive stage in life, especially in terms of mental health. It is estimated that 14% of young people between 10 and 19 years of age suffer from some type of mental disorder ([105]). The most frequent mental health disorders in this age group are anxiety and depression disorders (prevalence of 42.9%), conduct disorders (prevalence of 20.1%), attention and hyperactivity disorders (prevalence of 19.5%), intellectual and developmental disabilities (prevalence of 14.9%), and other mental disorders (prevalence of 9.5%) ([97]). Anxiety disorders and depression are the most prevalent during adolescence, followed by conduct disorders (attention deficit hyperactivity disorder (ADHD), oppositional defiant disorder (ODD), and dissocial disorder (DD)). Finally, suicide is another of the main mental health problems in adolescence; its prevalence has increased in recent years, mainly due to the economic and health crisis, and it is considered the fourth leading cause of death in this age group ([97]). Young people aged 15–29 years are considered to be one of the most vulnerable groups, with a higher prevalence in men (2:1) ([97]).

Authors such as [80] ([80]) report higher levels of depression, anxiety, stress and psychological discomfort, greater difficulty in expressing emotions, low self-control, low self-esteem, insomnia, and a greater likelihood of suffering psychological disorders as a result of internet abuse. Other authors ([30]) explain that these mental health problems derive from internet dependence, as internet abuse will lead to a loss of control, with withdrawal symptoms appearing and resulting in increased levels of anxiety, depression, and irritability. All this leads to a progressive increase in tolerance, increasing the time spent online with negative repercussions in all areas of their lives (personal, emotional, social, academic, and family).

Furthermore, problematic internet use also appears to be related to substance use ([90]). Evidence has been found of a direct relationship with tobacco use, but also with alcohol, cannabis, and other illegal drugs, but in these cases, consumption seemed to be mediated by tobacco use ([90]). Thus, internet addiction would constitute a risk factor for the development of other addictive behaviours ([90]). Finally, in relation to suicidal behaviours, authors such as [11] ([11]) considered that, although some elements of internet use may act as protective factors, in general, internet misuse is associated with an increased risk of suicidal ideation, self-harm, and depression.

Based on previous studies, some variables were identified as possible moderators that may contribute to inconsistency in the findings on associations between problematic internet use and mental health outcomes. Internet addiction is influenced by a combination of endogenous and exogenous factors. Endogenous factors include individual characteristics which may correlate positively or negatively with excessive device use ([95]). On the other hand, exogenous factors ([108]), such as demographic, cultural, or social factors, also play a crucial role, as they create an environment that facilitates or increases addiction to these devices.

There is evidence that being male or female may contribute to differences in this relationship. Some studies showed that the associations of problematic internet use with depressive symptoms ([99]; [65]), loneliness ([13]), and anxiety ([7]) were stronger for men than for women. However, findings to the contrary were also found ([57]), as well as findings that showed no differences between gender groups for other variables such as depression, anxiety, and subjective well-being ([13]).

Regarding age as a moderating variable, [57] ([57]) observed that the relationship between excessive internet use and subjective well-being, as well as positive emotions, was stronger in younger students compared to university students. This finding suggests that age may influence how internet use affects well-being, especially at earlier stages of development. In contrast, [41] ([41]) and [65] ([65]) found no significant results in their analysis.

Geographic region as a moderating variable has been shown to be significant in several analyses. [13] ([13]) found that study region acted as a statistical moderator for the observed heterogeneity in correlations between problematic internet use and depressive symptoms, loneliness, and other mental health outcomes. Specifically, these associations were stronger in studies conducted in South Asia and Europe, compared to other regions coded in their meta-analysis (Europe, West Asia, and East Asia). The same was true for the study by [57] ([57]), who observed that the link between negative emotions and excessive internet use was stronger in studies with participants from central and western China than in those with participants from eastern China. In contrast, [65] ([65]) found no significant evidence in their analysis. These findings highlight the importance of considering such variables as moderating factors in the present research.

Recent meta-analytic studies have found evidence of a relationship between problematic internet use and the appearance of internalizing symptomatology such as depression, anxiety, loneliness, self-esteem, or subjective well-being in adolescent or young adult populations ([65]; [13]; [57]; [41]; [26]) (see Table 1). [13] ([13]) found that higher levels of PIU were associated with higher levels of depression, anxiety, and loneliness and lower subjective well-being. [65] ([65]) focused on depressive disorder in adolescents aged 13–17, observing an increase in depressive disorder with higher internet use. [57] ([57]) identified that PIU is negatively correlated with subjective well-being, life satisfaction and positive emotions, and positively correlated with negative emotions. [41] ([41]) and [26] ([26]) revealed lower well-being when PIU scores were higher, assessing components such as self-esteem and life satisfaction.

As can be seen in Table 1, research on the relationship between internet addiction and externalizing problems in adolescents has been less investigated. In order to obtain more information about the role of these variables in internet addiction or problematic internet use, in addition to aggressiveness, externalizing symptomatology (externalizing problems, referring to general measures and impulsiveness) has been included.

This review examines the relationship between internet addiction/problematic internet use and mental health, taking into account both internalizing and externalizing symptomatology and focusing on the adolescent population. A meta-analytic approach was used to determine the impact of internet use on adolescent mental health, and two objectives were set: (a) to identify the relationship between inappropriate internet use (IA or PIU) and the most prevalent mental health variables in young people; (b) to quantify the strength of the relationship between each of them and the possible moderating variables (mean age, SD age, gender, JBI score, continent, and study design).

## 2. Materials and Methods

Following the objectives stated in this article, a literature review was carried out on the relationship between internet addiction and mental health to determine and quantify the impact of internet use on adolescent mental health.

The PRISMA guidelines for conducting and reporting systematic reviews and meta-analyses were followed. Appendix A includes the PRISMA guide checklist applied to our study ([79]). In addition, registration in the international systematic review database, PROSPERO, was carried out on 30 January 2024 ([76]).

### 2.1. Study Selection Criteria

Specific criteria were established for the inclusion of relevant articles in the analysis. These criteria included the selection of empirical and quantitative research articles, with a focus on the adolescent population aged 11–18 years, and written in English or Spanish. Articles that primarily addressed the relationship between addiction or problematic internet use and mental health were considered from any country of publication. In addition, the search was limited to articles published in the last five years, from 2019 to 2023, to ensure that the data are current and relevant to the constantly evolving field of internet use, ensuring that findings are applicable to the current context.

On the other hand, we excluded studies with a sample size of fewer than 30 participants, as well as those focused on video games or gaming, since the literature presented previously ([50]; [66]; [68]) highlights the difference between the concepts of GPIU and IGD and the existence of inconsistent results regarding the relationship between internet addiction or problematic internet use and online gaming. Thus, we decided to exclude these variables so as not to dilute the focus of this study. We also decided to exclude the COVID-19 variable from the search to avoid biased or unrepresentative results. Incorporating this variable could limit the validity of the results by mixing atypical contexts with more representative situations of the widespread use of social networks and their impact on well-being. In addition, articles whose main focus was not on internet use or which relegated internet use to a secondary role were discarded.

### 2.2. Search Strategy

To retrieve relevant studies in the field of internet addiction or problematic internet use and health, all the database collections of Web of Science (Clarivate Analytics, All Database) were used. These included bibliographic references of articles of scientific and academic interest in its core collection (Web of Science Core Collection) composed of eight databases: Medline, Current Contents Connect, SciELO Citation Index, Derwent Innovations Index, Grants Index, KCI-Korean Journal Database, Preprint Citation Index, and ProQuest Dissertations & Theses Citation Index.

The search was conducted in all Web of Science databases on 25 March 2023 and was limited to the last 5 years in the research areas of psychology, behavioural sciences, and pediatrics, with English and Spanish selected as the languages. This search was replicated by two more researchers.

The combination of terms entered into the search was as follows: ((((TS = (Internet addiction or excessive internet use or problematic internet use or digital technology use)) AND TS = (teenagers or adolescents or teens or youth)) AND TS = (mental health or mental illness or mental disorder or psychiatric illness or psychological consequences or psychological effect or psychological impact or psychological problems or psychological wellbeing or psychological well-being)) NOT TS = (gaming)) NOT TS = (COVID-19 or coronavirus or 2019-ncov or SARS-CoV-2 or CoV-19) NOT TS = (university students or college students or undergraduate students).

### 2.3. Study Selection

After the first search, as can be seen in the flow chart below (Figure 1), the total number of articles found was 830. Subsequently, after screening by title and abstract, the number decreased to 198 articles, of which, after a complete reading and application of the selection criteria, the number decreased to 68 articles.

Following an assessment of the methodological quality of these articles (according the JBI Critical Appraisal Checklist), 25 articles were eliminated for scoring less than 5 points on this scale, leaving 43 articles. These articles did not comply with some fundamental methodological aspects, such as the following: clear definition of the inclusion criteria in the sample, detailed description of the subjects and the environment, valid and reliable measurement of exposure, use of objective criteria to measure the condition, identification of confounding factors, proposal of strategies to address these factors, validity and reliability in measuring the results, and use of appropriate statistical analysis. Failure to meet these criteria limited the quality and reliability of the data presented in these studies, which could compromise the validity of the general conclusions. For this reason, we decided to exclude them to ensure greater methodological rigour and strengthen the evidence base used in this research.

Finally, 10 articles were eliminated because they did not provide sufficient quantitative information to calculate the effect size, leaving 33 articles.

### 2.4. Data Extraction from the Selected Studies and Coding of Variables

Once the final articles had been selected for review, a data extraction procedure based on variable coding followed. Each study was reviewed by two trained coders using a standard list of variables.

These were divided into the following variables: extrinsic, substantive, methodological, and outcome. The register carried out can be seen in Appendix A.

Extrinsic variables include data referring to the code of each article, authors’ citations, number of authors, year of publication, and the article’s quality score (total and percentage).Among the substantive variables we find those referring to subject and sample variables (sample size, percentage of women, mean and standard deviation of age and nationality).Methodological variables refer to design variables and variables related to internet addiction, depression, anxiety, stress, suicidal behaviour (referring to suicidal behaviour or self-harm), psychological well-being, externalizing problems, and internalizing problems (where descriptors and instruments are recorded).Finally, outcome variables refer to the statistic used and the numerical result of the relationship between internet addiction and depression, anxiety, stress, suicidal behaviour (referring to suicidal behaviour or self-harm), psychological well-being, self-esteem, body image, internalizing problems, externalizing problems, tobacco use, alcohol use, aggressiveness, impulsiveness, and delinquent behaviour.

### 2.5. Computation of Effect Sizes

The effect size index was the Pearson’s correlation coefficient calculated between mental health constructs and internet addiction. For each study, the Pearson’s correlations were transformed into Fisher’s Z metric to normalize distributions and stabilize variances. To make their interpretation easier, Fisher’s Z values for the individual effect sizes, the mean effect sizes and their confidence limits were back transformed into Pearson correlation metric ([10]).

Although most of the studies included reported the results in terms of correlations, some studies did not directly report correlation coefficients. For example, if a study reported the results by means of means and standard deviations or odds ratios, conversion formulas were applied to transform them into correlation coefficients (for more details see [91]). If the study reported the results by means of medians, approximation methods were applied to estimate the means and standard deviations (for more details see [100]).

Effect sizes were calculated for each mental health construct: depression, anxiety, stress, psychological well-being, self-esteem, externalizing problems, aggressiveness, impulsiveness, and suicidal behaviour. In all cases, positive correlations indicated a positive relationship between mental health and internet addiction, and negative correlations indicated a negative relationship between mental health and internet addiction.

### 2.6. Statistical Analyses

Separate meta-analyses were carried out to assess the relationship between each mental health construct (depression, anxiety, stress, psychological well-being, self-esteem, externalizing problems, aggressiveness, impulsiveness, suicidal behaviour) and internet addiction.

As variability was expected in effect sizes, random-effects models were assumed ([10]). These models involve weighting each effect sizes by its inverse variance, defined as the sum of the within-study variance and between-study variance, the latter being estimated by restricted maximum likelihood ([25]).

For each meta-analysis with more than 3 articles, a forest plot displaying the individual correlations was constructed, and the mean effect size with a 95% confidence interval was computed by means of the method proposed by [39] ([39]).

Cochran’s heterogeneity Q statistic and the I^2^ index were calculated to assess the heterogeneity among the effect sizes. The degree of heterogeneity was estimated using the I^2^ index (values of 0%, 25%, 50%, and 75%, representing no, low, moderate, and high heterogeneity, respectively) ([44]).

Publication bias was assessed using both a funnel plot with Duval and Tweedie’s trim-and-fill method for imputing missing data and the Egger test ([29]; [31]). A statistically significant result of the Egger test (*p* < 0.10) was evidence of publication bias. *p* < 0.10 was used instead of the usual *p* < 0.05 because of the lower statistical power of the Egger test with such a small number of studies.

High heterogeneity indicates that the differences in effect size observed across studies can be caused by the presence of moderating factors or other sources of variation. In this sense, if I^2^ > 90% and the number of studies was at least 10, analyses of potential moderator variables were performed ([1]).

Weighted ANOVAs and meta-regressions for categorical and continuous moderators, respectively, were applied by assuming mixed-effects models. An improved F statistic developed by Knapp and Hartung ([51]; [89]) was applied for testing the statistical significance of each moderator. In addition, an estimate of the proportion of variance accounted for by the moderator variable, R2, was calculated ([67]).

All statistical analyses were conducted with the metafor programme in R (4.2.3) ([98]).

## 3. Results

### 3.1. Assessment of Methodological Quality

To evaluate the quality of the articles selected in this meta-analysis, the tool “JBI Critical Appraisal Checklist”, more specifically one that evaluates cross-sectional studies, was used. This tool was developed by the JBI organization and its collaborators and approved by the JBI Scientific Committee after an extensive peer review ([74]). According to the organization, the objective of this assessment is to evaluate the methodological quality of a study and determine to what extent it has considered the possibility of bias in its design, execution and analysis. In addition, these authors stress the importance that during the conduct of a systematic review or meta-analysis, all articles selected for inclusion must undergo rigorous assessment by two critical appraisers.

In our study, this tool was used to assess the quality of the included studies (see Appendix A). Specifically, this tool includes eight items, which reflect different areas of study quality (sampling methods, inclusion criteria and sample description, validity and reliability, measurement tool, identification and strategies to address confounding factors, and appropriate use of statistical analysis). Each of the eight items was scored as “Yes”, “No”, “Unclear”, and “Not applicable” and assigned a score of “1” or “0”, with 1 being “Yes” and 0 being the other options. Therefore, the total score ranges from 0 to 8. The quality rating score was obtained by adding up the scores for each of the items on the scale. For this purpose, a total score was derived. Its values could be between 0 and 8, and the score was also obtained as a percentage (by dividing the total score by the total number of items multiplied by 100).

The quality assessment was carried out independently by two authors and discrepancies were resolved by discussion between the two assessors.

### 3.2. Characteristics of the Included Studies

The search strategy initially identified 830 potential articles for inclusion in this article. After the screening process, a total of 33 studies met the eligibility and methodological quality criteria and were selected for this meta-analysis (Figure 1). A summary of the main data and results of the selected articles is shown in Table A1 (Appendix B).

Of the 33 studies finally selected, published between 2019 and 2023, a total sample of 303,823 individuals was obtained. In this regard, it should be noted that the smallest sample was 60 and the largest was 154,981. In terms of gender, of the total sample, 50.56% were men and 49.44% were women. It should be noted that only one study did not include women among its participants, and only men were included in its sample. The mean age of the total sample was 14.57 years; only two articles did not provide a mean age or standard deviation but gave an age range, while the remaining articles reported the mean age of their participants.

In terms of cultural differences, 66.67% of the selected studies were conducted in East Asia (China, Japan, Taiwan, and Macau), with the East Asian population accounting for 46.2% of the total sample. Similarly, 9.09% of the studies were conducted in West Asia (Lebanon, Israel, and Turkey) and another 9.09% in Europe (Spain and Slovakia), accounting for 0.74% and 1.55% of the total sample, respectively. A total of 6.06% of the research was carried out in South Asia (India and Malaysia) and another 6.06% in South America (Brazil), accounting for 0.14% and 0.37% of the total sample, respectively. Finally, there was one study, representing 3.03% of the total number of articles, where the participants’ nationality was shared between Europe, North America, and West Asia, accounting for 51.01% of participants in the total sample. The presence of different countries allows us to see whether there are differences between cultures.

The most frequent study design was cross-sectional (n = 28, 84.84%), with the rest of the studies being longitudinal (n = 5, 15.15%). Most of the studies did not report the time period assessed, while eight studies conducted their study over a period of 4–9 months; in one study, the time period was 1 year, in two studies, the time period was 2 years, and in one study, the time period was 4 years.

In terms of study outcomes, 18 studies assessed depression (41,693 subjects), 7 studies assessed anxiety (27,122 subjects), 2 studies assessed stress (8456 subjects), 6 studies assessed suicidal behaviour (82,872 subjects), 8 studies assessed psychological well-being (164,918 subjects), 5 studies assessed self-esteem (4742 subjects), 2 studies assessed externalizing problems (3172 subjects), 3 studies assessed aggressiveness (29,587 subjects), and 2 studies assessed impulsiveness (13,610 subjects).

The instruments used for each variable can be consulted in Table A2 (Appendix C), which highlights that the most widely used variable to measure internet addiction was the IAT. As for the psychological variables, the CES-D was most frequently used for depression and the GAD-7 and the DASS-21 for anxiety; the latter was also the most used for stress and psychological well-being. For suicidal behaviour, questionnaires proposed by the authors were used most often; for self-esteem, the RSES; for aggressiveness, the AQ was the most used; for impulsiveness, the only instrument found was the BISS-11; and for externalizing problems, the YSR and the SDQ were most frequent.

### 3.3. Mean Correlations and Heterogeneity

Table 2 shows the results of the nine meta-analyses performed for each outcome. For most of the variables analyzed, the results were statistically significant, as can be observed in the variables where the intervals do not cross the null value. The highest mean correlation was found for the relationship between internet addiction and aggressiveness (*r*+ = 0.391; 95% CI = 0.244, 0.521), followed by the mean correlation for internet addiction and depression (*r*+ = 0.318; 95% CI = 0.214, 0.415). Similar mean correlations were found for psychological well-being (*r*+ = −0.312; 95% CI = −0.407, −0.212) and self-esteem (*r*+ = −0.306; 95% CI = −0.527, −0.047). Last, the mean correlations that estimated the relationship between internet addiction and anxiety and between internet addiction and suicidal behaviour were the smallest (*r*+ = 0.252; 95% CI = 0.078, 0.412 and *r*+ = 0.264; 95% CI = 0.185, 0.339, respectively). In all cases, the mean correlations reflected moderate magnitudes. Furthermore, the sign of the correlations indicated the sense of the relationship between internet addiction and each outcome. However, some results were not statistically significant, as seen in the correlations for externalizing problems (general measures) (*r*+ = 0.292; 95% CI = −0.487, 0.813), impulsiveness (*r*+ = 0.303; 95% CI = −0.605, 0.868), and stress (*r*+ = 0.253; 95% CI = −0.996, 0.999), probably due to the low statistical power of such a small number of studies (*k* = 2).

It is important to consider the scarce number of studies that assessed the relationship between internet addiction and externalizing problems, aggressiveness, impulsiveness, and stress. For this reason, these results can be interpreted cautiously.

Finally, as can be seen in Table 2, the analyses found considerable heterogeneity among individual correlations in all meta-analyses (*I*^2^ > 90% and *p* < 0.001). Figure 2 presents the forest plots of the studies with more than three articles. All the relevant results of the quantitative synthesis are summarized in it.

### 3.4. Publication Bias

Publication bias was analyzed for depression, anxiety, psychological well-being, self-esteem, and suicidal behaviour outcomes by constructed a funnel plot, and its asymmetry was assessed with the trim-and-fill method and the Egger test.

Non-significant results for the interception were obtained from the Egger test for depression (t(16) = −0.935; *p* = 0.364), anxiety (t(5) = −1.285; *p* = 0.255), psychological well-being (t(6) = 0.474; *p* = 0.652), and suicidal behaviour (t(4) = −2.271; *p* = 0.105). In addition, applying the trim-and-fill method, no correlation coefficients had to be imputed to achieve the symmetry of the funnel plots for depression, anxiety, psychological well-being, and suicidal behaviour (see Figure 3A, Figure 3B, Figure 3C, Figure 3D and Figure 3E, respectively). These results led us to discard publication bias as a threat against these meta-analytic results.

For self-esteem outcomes, the Egger test yielded a significant result (*p* < 0.10) for interception (t(3) = −3.067; *p* = 0.053), and by applying the trim-and-fill method, an additional correlation coefficient was imputed to the set of original correlations to achieve symmetry in the funnel plot (see Figure 3D). The adjusted mean correlation, once corrected by publication bias, was radj = −0.355 (95% CI = −0.513, −0.196; *k* = 6). Compared with the original mean correlation obtained for the five studies (*r*+ = −0.306), the adjusted mean correlation barely changed.

### 3.5. Analyses of Moderators

The heterogeneity found among the correlations led us to carry out analyses of moderator variables for the outcome with at least 10 studies, i.e., depression.

Table 3 presents the results of the simple meta-regressions performed on continuous moderators. None of the analyzed moderators reached a statistically significant association (*p* < 0.05) with the correlations between internet addiction and depression.

The continent where the study took place and study design of the were also analysed as categorical moderators by means of weighted ANOVAs models. Table 4 presents those results. Once again, none of them reached a statistically significant association (*F*_2,15_ = 0.53, *p* = 0.601 and *F*_1,16_ = 0.16, *p* = 0.697, respectively).

## 4. Discussion

Our objective was to determine the impact of internet use on adolescents’ mental health. To this end, a meta-analytic study was conducted on the relationship between the most prevalent mental health variables and internet addiction, quantifying the strength of this relationship and possible moderating variables, such as age, sex, quality of the selected articles, nationality of the studies, and type of design used. We estimated the associations between problematic internet use and the nine mental health variables analyzed in the selected studies: depression, anxiety, stress, psychological well-being, self-esteem, suicidal behaviour, externalizing problems, aggressiveness, and impulsiveness.

The results indicate that internet addiction is related to the variables depression, anxiety, mental well-being, self-esteem, aggressiveness, and suicidal behaviour. The strength of these relationships reflects moderate magnitudes, with the relationship between internet addiction and aggressiveness being the highest, followed by the relationship with depression. These are followed by the relationship between internet addiction and psychological well-being and self-esteem, respectively. The lowest relationship was found between internet addiction and the variables anxiety and suicidal behaviour. However, no significant relationships were found between internet addiction and the variables of stress, impulsiveness, and externalizing problems (general measures), probably due to the small number of articles that analyzed the relationship with these variables.

Regarding the influence of moderating variables (mean age of the sample, standard deviation of age, percentage of women, study quality score (JBI), nationality, and study design), only their possible effect on the relationship between depression and internet addiction could be estimated. The results showed that these moderating variables had no effect on this relationship, indicating that this connection between internet addiction and depression was robust with regard these variables. The absence of significant findings for these moderators could be explained by the methodological diversity among the reviewed studies, including differences in the diagnostic criteria used to identify IA/PIU, the characteristics of the selected samples, and the statistical approaches used. Cultural factors also probably play a crucial role, as in Asian countries, the prevalence of IA/PIU is higher than in other regions.

Aggressiveness has been identified as one of the most prevalent externalizing behaviours in adolescence in the study of psychopathology ([47]). In our meta-analytic study, it is one of the behaviours most strongly related to internet addiction. [77] ([77]) and [81] ([81]) found that adolescents with higher scores on internet addiction showed higher levels of aggressiveness. [81] ([81]), in addition to supporting the idea that internet addiction was a positive predictor of aggression, showed that aggression was at the same time a predictor of internet addiction, both being a risk factor for the other. On the other hand, [42] ([42]), conducted a study where they observed the effect that an aggressive personality has on problematic internet use and suicidal ideation in adolescents, finding that participants who have more problems with the internet show greater suicidal ideation, with this association being stronger in groups with aggressiveness. These results are consistent with previous research showing that both the frequency of use and the severity of internet addiction are related to higher aggression scores ([49]; [53]). In this sense, internet use seems to facilitate the expression of latent aggressive impulses, such as repressed anger, aggression, or hostility, which are not acceptable in society but can be released in the digital environment; the relief of these emotional states generates a rewarding feeling that seems to lead to addiction ([53]; [33]).

Several studies highlight the association between depression and internet addiction ([77]; [20]; [106]; [22]; [34]). Only one of the studies included in this meta-analysis, [3] ([3]), found no significant associations between problematic internet use and emotional variables, including depression. The authors claim that this lack of association could be explained by the small sample size. Among those supporting this association is the study by [32] ([32]), who revealed how adolescents with PIU show greater depressive symptoms compared to patients without PIU. The study by [101] ([101]) with adolescents with a diagnosis of Major Depressive Disorder concluded that adolescents with more severe depressive symptoms are more likely to have symptoms of internet addiction. [84] ([84]) showed an association of problematic smartphone use (PSU) with depressive symptoms, and the results suggest that adolescents with PSU are more likely to experience mood disturbances and depressive symptoms than adolescents without PSU. [111] ([111]) found a direct relationship between problematic social media use, more specifically Instagram, with depression and other psychopathological symptoms.

Regarding the direction of the relationship between depression and internet addiction among the selected studies, the longitudinal studies by [110] ([110]) and [58] ([58]) or the cross-sectional study by [34] ([34]) stated that it is depressive symptoms that predict the onset of internet addiction, while others ([42]; [52]) found that problematic internet use precedes depressive symptoms.

In this sense, one of the most common explanatory hypotheses for this relationship is displacement theory, which posits that spending excessive time on the internet or using digital devices reduces face-to-face interactions and takes time away from other important tasks. This can deteriorate real-life interpersonal relationships, leading to feelings of isolation and psychological difficulties, such as anxiety, depression, and low self-esteem, ultimately impacting the individual’s mental health negatively ([7]; [96]).

On the other hand, some authors suggest the reverse hypothesis, that people with depressive symptoms use the internet to reduce negative feelings, a behaviour described by some authors as compensatory internet use. This concept attempts to explain the frequent assumption that individuals turn to the internet to escape real-life problems or alleviate dysphoric moods, a practice that can sometimes lead to negative outcomes ([49]; [46]). In the same way, research by [40] ([40]) and [88] ([88]) found that depression led to increased social media use. [19] ([19]) show that depression predicts and supports the persistence of young people’s internet addiction. However, [8] ([8]) claim that it is internet addiction that affects depressive symptoms, proving that levels of depression can be reduced if internet use is controlled through household rules. These results suggest a bidirectional relationship indicating that while internet addiction may exacerbate depressive symptoms, underlying mental health conditions may also drive problematic internet use, creating a cycle that worsens both.

The mediating role of depression in the relationship between IA/PIU and other variables, such as perception of school climate ([112]), bullying victimization ([63]), or peer victimization ([60]), has also been analyzed. Depression proved to be a mediating variable in these relationships, determining that the relationship between depression and IA or PIU was significant and positive. Similarly, a longitudinal study by [14] ([14]) examines the mediating effect of IA concerning the relationship between depression and bullying victimization, identifying IA as a mediating variable and establishing a significant positive relationship between depression and IA. Previous studies confirm this relationship, indicating that higher levels of internet addiction are related to higher levels of depression ([13]; [23]; [61]; [2]; [36]).

Regarding psychological well-being, some studies support its association with internet addiction ([48]; [102]; [87]), noting that higher levels of internet addiction are associated with poorer levels of psychological well-being. [102] ([102]) report a positive association between PIU and behavioural and emotional problems, while Pontes and Macur’s study (2021) finds that PIU is associated with lower levels of subjective well-being.

The longitudinal study by [42] ([42]) identified three distinct trajectories of IA (low increase, moderate decrease, and high decrease) among adolescents over three years. Their results indicated that being in the high or moderate IA risk groups was associated with lower psychological well-being. [4] ([4]) found a direct association between problematic smartphone use (PSU) and psychological distress. [9] ([9]) analyzed the problematic social network use (PSMU) and psychological well-being of adolescents with data from 29 countries and found that PSMU is indicative of low well-being in all domains assessed and in all countries.

Psychological well-being was also analyzed as a mediating variable in the relationship between PIU and other behavioural or mental health variables. [64] ([64]) focused on the mediating role of negative affectivity in the relationship between PIU and suicidality and self-injurious behaviours; [109] ([109]) analyzed the mediating role of mental health in the relationship between PIU and externalizing behavioural problems among adolescents. In both cases, psychological well-being was found to be a mediating variable between these relationships, as well as indicating a positive association between these variables, determining that the higher the PIU, the worse the psychological well-being.

Thus, according to the authors reviewed in this meta-analytic study, it is generally observed that higher levels of addiction are associated with poorer psychological well-being. Previous studies ([62]; [72]) agree with this result. According to [17] ([17]), an explanatory hypothesis is that the variable psychological well-being is related to a mental state characterized by feelings such as satisfaction, pleasure, self-perception of well-being, and positive emotions, which contrasts with aspects usually linked to internet addiction, such as loneliness, poorer social adaptation, poorer emotional skills, and neuroticism, all of which hinder the individual’s adaptation and coping ability.

As with psychological well-being, self-esteem was also inversely related to internet addiction. Studies by [77] ([77]) and [20] ([20]) support the association between self-esteem and internet addiction, indicating that higher levels of internet addiction resulted in lower levels of self-esteem. Also, as with psychological well-being in the longitudinal study by [42] ([42]), the results indicate that being in the high or moderate IA risk groups is associated with lower self-esteem. In the research carried out by [71] ([71]), they observe that higher levels of PIU resulted in lower levels of self-esteem.

Finally, a study by [94] ([94]) focuses on the mediating role of body self-esteem in the relationship between IA and sexual victimization in adolescents. It was found that self-esteem did affect this relationship, as well as being associated with IA, determining that the higher the IA, the lower the self-esteem. In support of these results, previous studies ([17]; [5]) agree with these findings and found similar results. [35] ([35]) explains that individuals who evaluate themselves negatively may perceive the internet as a means to compensate for some self-perceived shortcomings or negative aspects. Through anonymity or the absence of face-to-face contact, the internet helps them with the adoption of another personality or social identity, thus satisfying these shortcomings and generating a dependent relationship with this medium.

In relation to anxiety and its association with IA, several studies support the relationship that higher levels of anxiety lead to higher levels of internet addiction ([77]; [34]; [58]). These include [32] ([32]), who in their study reveal how adolescents with PIU showed higher anxious symptoms compared to patients without PIU. However, [3] ([3]), as with depression, did not find significant associations between PIU and anxiety due to the small sample used. [111] ([111]) found that problematic social media use (PSMU), more specifically Instagram, is directly positively associated with anxiety for the male sample, while for females, this association was indirect. The study by [60] ([60]) focused on the mediating role of anxiety on the basis of the relationship between IA and peer victimization, determining how anxiety affects this relationship, as well as observing that higher levels of IA lead to higher levels of anxiety. Previous studies confirm these results ([2]; [104]). This fact could be explained, on the one hand, by the fact that people with anxiety have a preference for online social interactions as these interactions have benefits for these people; they help them to maintain control over some variables such as the speed of interaction, the elaboration of sentences, or the way in which they appear in front of the other person, factors that help to increase feelings of security and confidence ([2]; [15]). On the other hand, instances when internet access is restricted, when messages cannot be responded to immediately, or when the FOMO phenomenon occurs have also been found to be associated with increased symptoms of anxiety ([104]).

Regarding the relationship between IA and suicidal and self-harming behaviours, some studies found that increases in IA scores led to increases in suicidal behaviours, including suicidal ideation (SI), suicidal plans (SPs), and suicidal attempts (SAs) ([82], [83]). In relation to suicidal ideation, other studies ([54]; [43]) agree that the higher the IA/PIU, the higher the suicidal thoughts. Furthermore, the study by [64] ([64]) found a direct effect of PIU on suicidality and self-injurious behaviours (SSIBs) and the study by [37] ([37]) found that an increase in IA/PIU was related to suicidal and self-injurious behaviours. Suicidal behaviours, especially in young people, are one of the problems with the greatest impact on global public health ([97]). It is therefore important to look for variables that may be related or may be a risk factor for these behaviours to take place.

Previous studies agree with the results described above ([11]; [21]; [69]; [45]). However, authors such as [27] ([27]) claim that internet use can have protective effects as it can serve as a source of emotional support and improve coping strategies. Nevertheless, when talking about problematic internet use, there seem to be more negative effects in relation to an increased risk of self-harm or suicidal ideation, as inappropriate internet use facilitates exposure to suicidal behaviour and is associated with more dangerous methods of self-harm ([11]).

Finally, regarding the variables of stress, impulsivity, and externalizing problems (general measures), our study does not yield significant results due to the small number of articles found for this meta-analysis in relation to these variables. We will now briefly comment on the most significant results of the analyzed studies.

In relation to stress and its association with IA, [34] ([34]) support the relationship that higher levels of stress lead to higher levels of internet addiction. However, in the study by [3] ([3]), as with depression and anxiety, no significant associations are found between PIU and stress, alleging that this could be explained by the small sample used.

In relation to impulsivity, [77] ([77]) find more internet addiction in students with higher levels of impulsivity, and [42] ([42]) report a stronger association between PIU and suicidal ideation in students with impulsive personality.

For externalizing problems (in relation to general measures), [109] ([109]) find that PIU plays a negative role in promoting externalizing problem behaviours, as a positive association is found between these two variables. As in the study by [102] ([102]) they point out that the higher the PIU levels, the higher the presence of behavioural problems, especially in relation to hyperactivity and conduct problems.

### Limitations

This paper included studies from around the world and illustrated the relationship between various mental health variables and AI, including symptomatological variables characteristic of both internalizing and externalizing disorders and positive variables such as well-being and self-esteem. We included only validated measures of IA for each of the nine mental health variables, and studies with a quality of between 5 and 8 points on the JBI scale, eliminating those below 5 points. However, our study also has limitations.

Firstly, the number of articles found for each of the mental health variables, with the exception of studies on depression, was below 10 articles. In this sense, this small number of articles for most of the variables only allowed us to take into account moderating variables in the relationship between depression and internet addiction. At the same time, the absence of significant findings in this relationship may be due to the methodological heterogeneity of the selected studies, as well as the possible influence of cultural factors not considered. There is also a potential bias related to the sample selected, given that a significant portion of the included studies come from East Asian regions, where cultural, technological, and social patterns may differ from other geographies. This could limit the generalizability of the findings to global contexts.

Secondly, some of the variables found after data extraction had to be extracted from the analyses due to the scarcity of studies found (n = 1) (body image, internalizing problems, delinquent behaviour) and the impossibility of extracting the effect size from the original item (tobacco use and alcohol use).

Third, although some studies included in this meta-analysis explored the bidirectional nature of some of the associations analyzed, most of them had a cross-sectional design, which limits the possibility of making inferences about the causal relationship between internet addiction and adolescent mental health. Therefore, future studies can employ longitudinal designs to further explore this bidirectionality and shed more light on these dynamics.

Fourthly, no publication biases were found for any variable except self-esteem, which forces us to interpret the data with caution with respect to this variable.

Finally, the missing data could limit the interpretation of moderator analyses.

## 5. Conclusions

Due to the increasing use of the internet, several studies have focused on the relationship between internet addiction and mental health-related problems. The present meta-analysis aims to synthesize and quantify the evidence found in recent years on the relationship of IA with mental health problems, both internalizing and externalizing, in the adolescent population.

The selected studies found evidence of a positive association between IA and variables such as depression, anxiety, stress, suicidal behaviour, externalizing problems, aggressiveness, and impulsiveness, as well as a negative association with self-esteem and psychological well-being. Only one study ([3]) was unable to determine the existence of a relationship between IA and depression, anxiety, and stress.

The results of our meta-analysis confirm all relationships, except for the variables of stress, impulsivity, and externalizing problems (general measures), where no significant associations with AI were found. The strength of the relationships found is of moderate magnitude and is the strongest for aggression and depression and the weakest for anxiety and suicidal behaviour.

These results help to expand our knowledge about internet addiction and its relationship with mental health variables in young people, emphasizing not only internalizing variables but also externalizing variables such as aggression, for which we have less evidence. The study of the consequences of problematic internet use on mental health is a fundamental requirement for making progress in the design and implementation of more effective and efficient interventions in a population as vulnerable as adolescents.

## Figures and Tables

**Figure 1 behavsci-15-00116-f001:**
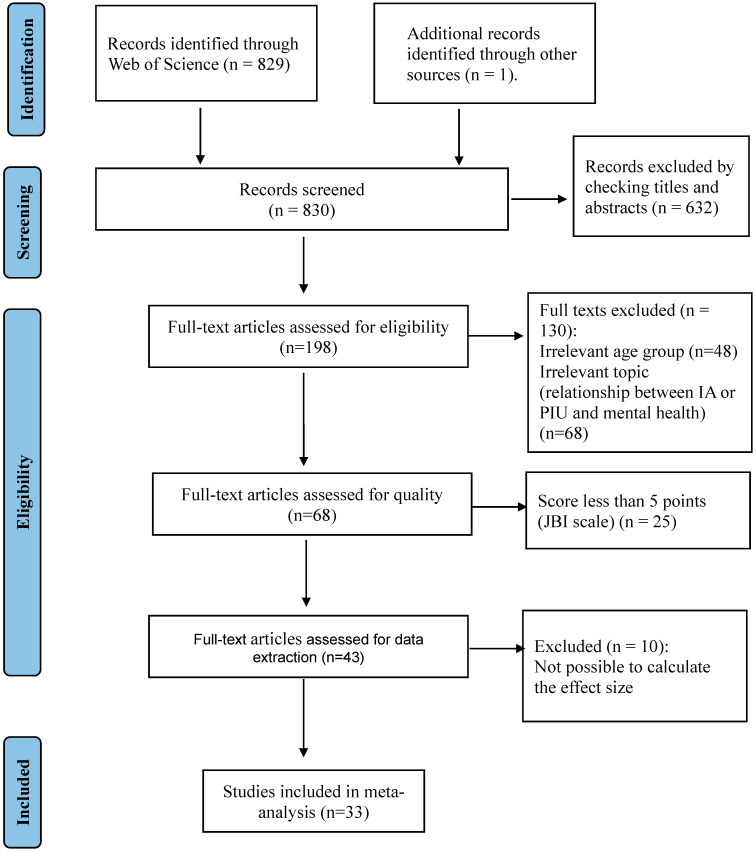
PRISMA flow diagram of the selection of studies for the meta-analysis.

**Figure 2 behavsci-15-00116-f002:**
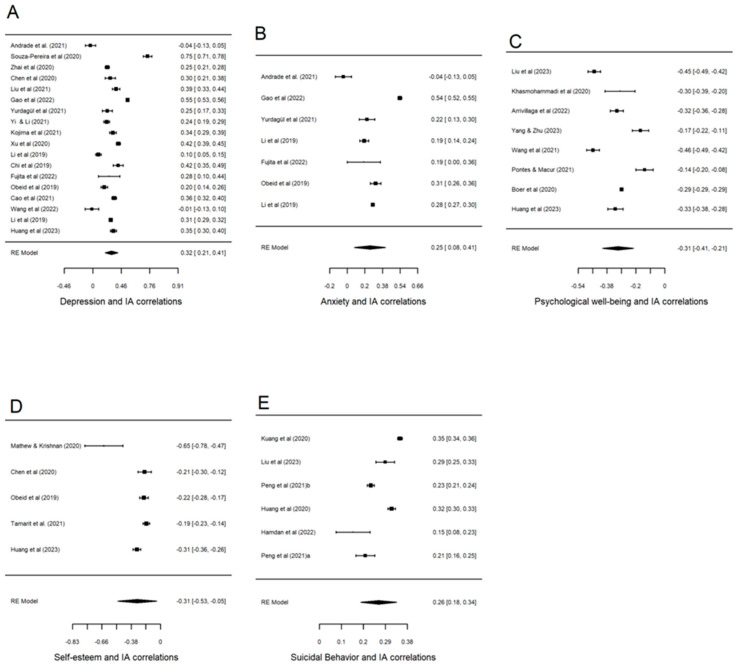
Forest plots displaying correlations with 95% confidence intervals between internet addiction and depression (**A**), anxiety (**B**), psychological well-being (**C**), self-esteem (**D**), and suicidal behaviour (**E**). RE (random-effects) model refers to the statistical model assumed in the computation of the mean correlation coefficient ([3]; [84]; [112]; [20]; [63]; [34]; [111]; [110]; [52]; [106]; [60]; [58]; [22]; [32]; [77]; [14]; [101]; [42]; [48]; [109]; [87]; [9]; [71]; [94]; [54]; [64]; [82]; [83]; [37]).

**Figure 3 behavsci-15-00116-f003:**
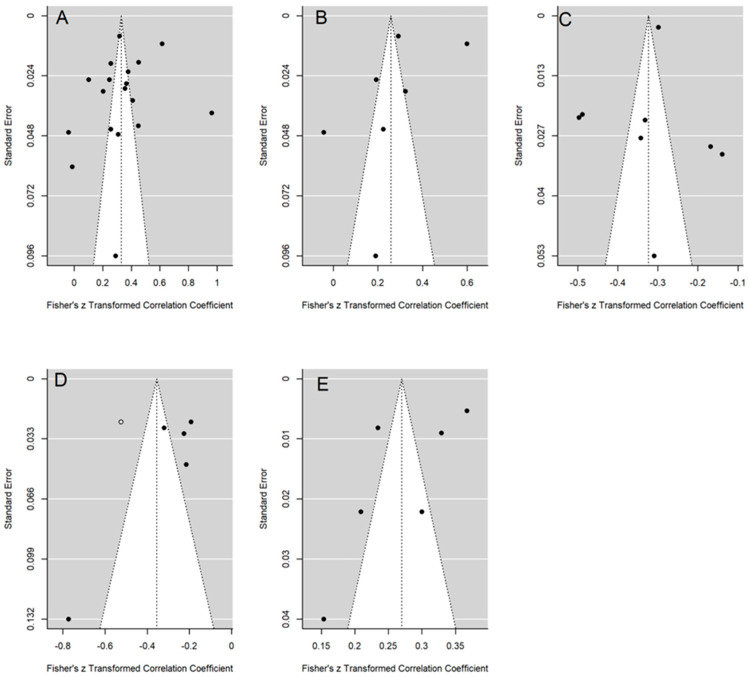
Funnel plots of depression (**A**), anxiety (**B**), psychological well-being (**C**), self-esteem (**D**), and suicidal behaviour (**E**). The white circle is the correlation coefficient imputed by means of the trim-and-fill method.

**Table 1 behavsci-15-00116-t001:** Meta-analyses on the relationships between problematic internet use and mental health.

Study	Study Count	Age Group	Internalizing Outcomes	Externalizing Outcomes	Moderator Analyses	Results
[13] ([13])	223	Students from elementary, middle, high school, and colleges	Depressive symptoms (DSs), anxiety (A), loneliness (L), subjective well-being (SWB)	“Other mental health outcomes” (suicide, aggression, and hostility) Unable to investigate the relationship with PIU	School grade, region, measure of PIU, publication year, gender	*rDS* = 0.313 *rA* = 0.277 *rL* = 0.252 *rSWB* = −0.213
[65] ([65])	13	Adolescents 13–17 years	Depressive disorder	None	Sex, mean age, culture	*r* = 0.3
[57] ([57])	70	Youth and college students	Subjective well-being (SWB), life satisfaction (LS), positive emotion (PE), negative emotion (NE)	None	Region, age, gender	*rSWB* = −0.313 *rLS* = −0.21 *rPE* = −0.183 *rNE*= 0.251
[41] ([41])	40	All age groups	Psychological well-being	None	Type of internet use, indicator of well-being, quality of internet use measure, age, gender	*R* = −0.0385
[26] ([26])	23	All age groups	Well-being	None	Well-being components (self-esteem, well-being, and life satisfaction)	*r* = −0.18

**Table 2 behavsci-15-00116-t002:** Mean correlations between internet addiction and each outcome, 95% confidence intervals, and heterogeneity statistics.

Outcome	*k*	*r*+	95% CI	*Q*	*p*	*I* ^2^
			LL UL			
Depression	18	0.318	[0.214, 0.415]	1104.56	<0.001	98.98
Anxiety	7	0.252	[0.078, 0.412]	656.31	<0.001	98.98
Psychological well-being	8	−0.312	[−0.407, −0.212]	203.04	<0.001	97.78
Self-esteem	5	−0.306	[−0.527, −0.047]	29.36	<0.001	96.63
Suicidal behaviour	6	0.264	[0.185, 0.339]	237.17	<0.001	98.02
Externalizing problems	2	0.292	[−0.487, 0.813]	12.81	<0.001	92.19
Aggressiveness	3	0.391	[0.244, 0.521]	38.84	<0.001	96.82
Impulsiveness	2	0.303	[−0.605, 0.868]	25.72	<0.001	96.11
Stress	2	0.253	[−0.996, 0.999]	124.97	<0.001	99.20

*k* = number of studies; *r*+ = mean correlation coefficient. LL and UL: lower and upper limits of the 95% confidence interval for *r*_+._
*Q* = Cochran’s heterogeneity *Q* statistic; *Q* statistic has *k* − 1 degrees of freedom. *p* = probability level for the *Q* statistic. *I*^2^ = heterogeneity index.

**Table 3 behavsci-15-00116-t003:** Results of meta-regressions for the influence of continuous moderators on the correlations between internet addiction and depression.

	*k*	*b* _j_	*F*	*p*	*Q* _E_	*p*	*R* ^2^
Mean age	16	0.015	0.10	0.754	545.16	<0.001	0%
SD age	16	−0.048	0.13	0.719	550.15	<0.001	0%
Gender (% of women)	18	−0.013	3.62	0.075	901.58	<0.001	13.99%
JBI score	18	0.114	1.84	0.194	1054.89	<0.001	4.64%

*k* = number of studies. *b*_j_ = regression coefficient. *F* = *F* statistic to test the statistical significance of the moderator. *Q*_E_ = Statistic for testing the model misspecification. *R*^2^ = percentage of variance accounted for by the moderator. *p* = probability level. JBI score = total score obtained in the JBI Critical Appraisal Checklist to assess the study quality.

**Table 4 behavsci-15-00116-t004:** Results of the weighted ANOVAs for the influence of categorical moderators on the correlations between internet addiction and depression.

	*k*	*r*+	95% CI	ANOVA Results
			LL LU	
Continent:				
East Asia	14	0.313	[0.189, 0.427]	*F*_2,15_ = 0.53, *p* = 0.601
West Asia	2	0.224	[−0.120, 0.521]	*R*^2^ = 0
South America	2	0.433	[0.113, 0.672]	*Q*_W_(15) = 1030.66, *p* < 0.001
Study design:				*F*_1,16_ = 0.16, *p* = 0.697
Cross-sectional	13	0.329	[0.203, 0.445]	*R*^2^ = 0
Longitudinal	5	0.286	[0.077, 0.472]	*Q*_W_(16) = 1027.50, *p* < 0.001

*k* = number of studies. *r*+ = mean correlation coefficient. LL and UL: lower and upper limits of the 95% confidence interval (95% CI) for *r*+. *F* = *F* statistic for testing the statistical significance of the moderator. *R*^2^ = percentage of variance accounted for by the moderator. *Q*_w_ = statistic for testing the model misspecification.

## Data Availability

Data on guidelines for the publication of systematic reviews and/or meta-analyses and structured abstracts, coding of selected variables for analysis in SPSS, and quality checklists of studies are available in the Appendix A of this article.

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
