# Peer review of "The Association Between Internet Addiction and Adolescents’ Mental Health: A Meta-Analytic Review"

_behavsci, 2025, doi:10.3390/bs15020116_

Round 1

Reviewer 1 Report

Comments and Suggestions for Authors

Problematic factors of internet use impact on the mental outcomes is the focus of the study. A meta analysis is done. The number of papers is not impressive. The author eliminated many potnetial papers with their search. For instance how was covid eliminated. Even though the studies were performed in the times of covid, maybe covid was not a direct impact of internet use. Particularly it was a context in whihc the impact of excessive use of internet could be seen on mental health, for example Identity Disturbance in the Digital Era during the COVID-19 Pandemic: The Adverse Effects of Social Media and Job Stress. So the paper should calrify those criteria of inclusion, and then address it in limitations, and future studies.

Additionally, the intro can analyze the paper such as  Graduate socialization and anxiety: insights via hierarchical regression analysis and beyond. Studies in Higher Education, and  Research on the Influencing Factors and Mechanism of Smartphone Use and Addiction on Employees: A Systematic Review. Journal of Chinese Human Resources Management, 

Both introduction and the discussion need better streamlined and structured literature review mentioned, with more natural flow, and coherence rather than listing what was done. 

The problematic internet use, is also not that well defined. There should also be some distinction between social media, mobile and desktop use, and type of content of consumed. The topic is quite wide , and sample of papers heterogenous in terms of the measurments and variables to draw specific conclusions

Author Response

Thank you very much for taking the time to review this manuscript. Please see the attachment where you can find the detailed responses in the attached document and the corresponding revisions/corrections highlighted/in track changes in the re-submitted files.

Reviewer 2 Report

Comments and Suggestions for Authors

The focus on Internet addiction (IA) among adolescents and its impact on mental health is timely and critical given its increasing prevalence.

The meta-analytic approach adds robustness to the findings and allows for comprehensive insight.

The manuscript is well-structured, with a clear flow from introduction to conclusions. However, some sections could benefit from additional elaboration, particularly on methodological decisions. The study offers valuable correlations between IA and mental health outcomes such as depression, anxiety, and aggression. These findings have implications for both academic research and public health interventions.

While the introduction provides a strong rationale for the study, incorporating more recent statistics or global trends in IA prevalence might strengthen the argument.

References to broader theoretical frameworks on digital behavior in adolescents would enhance the contextual background (use for example https://doi.org/10.3390/ijerph182111382 and doi: 10.2196/17341)

The inclusion and exclusion criteria are well-defined but would benefit from more justification. For instance, why were studies on gaming excluded, given its overlap with IA?

The PRISMA diagram is clear, but including a brief discussion on why some articles failed the JBI criteria might provide additional transparency.

The reliance on random-effects models is appropriate given the expected heterogeneity. However, a deeper explanation of how missing data were handled in moderator analyses could improve clarity.

Including examples of conversion formulas for effect size transformations (e.g., from medians to Pearson's correlation) would strengthen methodological rigor.

The tables and figures are informative but could benefit from additional annotations or brief descriptions within the text to ensure interpretability.

The discussion of heterogeneity (I² > 90%) is crucial but could elaborate on its implications for interpreting the meta-analytic results.

The lack of significant findings for moderators like gender and region warrants further discussion. Could methodological diversity or cultural factors contribute to these null results?

The discussion effectively highlights key relationships (e.g., IA and aggression, IA and depression). Still, it could delve deeper into potential mechanisms, such as the role of social media platforms in exacerbating IA symptoms.

The bidirectional nature of some associations (e.g., IA leading to depression vs. depression leading to IA) is acknowledged but could be further explored. While limitations are well-articulated, acknowledging potential biases in the selected sample (e.g., geographic concentration in East Asia) would be beneficial.

Consider suggesting future studies focusing on longitudinal designs to clarify causal relationships.

Author Response

(The authors gave the same response as above.)

Reviewer 3 Report

Comments and Suggestions for Authors

This is a very timely meta-analytic review examining the relationship between internet addiction and various mental health outcomes in adolescent. The meta-analysis was well-conducted and the findings are interesting. I agree that the paper will have a good contribution to the literature. I have several comments to improve the manuscript further:

1. First, I think some parts of the writing can be improved. The document does not consistently follow a clear indentation style. For example, some paragraphs are aligned to the left margin, while others appear slightly indented. Maintaining a consistent indentation style throughout the paper improves readability and professionalism.

2. Also, some paragraphs are overly long, making it harder to follow the flow of ideas. For example. the introduction contains long blocks of text that discuss historical context, operational definitions, and research gaps all in one paragraph. Breaking these into smaller, focused paragraphs would improve comprehension. These paragraph should be accompanied with strong topic sentence.

3. The discussion could benefit from greater theoretical depth. Integrating established frameworks, such as the compensatory Internet use model, the uses and gratifications theory or displacement theory, would provide a more nuanced interpretation of the findings. See the following papers on possible theoretical framework: A conceptual and methodological critique of internet addiction research: Towards a model of compensatory internet use. (2014) Computers in human behavior, 31, 351-354. and Why people use social media: a uses and gratifications approach. (2013) Qualitative market research: an international journal, 16(4), 362-369. and More time on technology, less happiness? Associations between digital-media use and psychological well-being. (2019). Current Directions in Psychological Science, 28(4), 372-379.

4. Furthermore, the paper frequently implies causality in relationships (e.g., Internet addiction causing depression or anxiety). Given the predominance of cross-sectional studies (84.84%), such causal claims are not supported by the data. Stronger emphasis on the limitations of causal interpretation is necessary. Specifically, I would like the authors to have a more elaborated discussion and acknowledgement on the possibility of reverse causation where mental health issues, such as depression and anxiety, are antecedent rather than outcome of internet/social media addiction.  This perspective is supported by prior research indicating that individuals with poor mental health engage in social media as coping mechanism. The issue regarding reverse causation has been well-discussed in the literature and should be highlighted in the current study to provide a more balanced perspective. Please see the following paper: Does social media use increase depressive symptoms? A reverse causation perspective. (2024) Frontiers in Psychiatry, 12, 641934.

5. There should be a justification on why the authors are limited the search to the last 5 years

6. Heterogeneity values for key outcomes are quite high. There should be more discussion on potential sources of heterogeneity (e.g., measurement tools, cultural differences, study design variations).

7. The exclusion of studies focusing on COVID-19 appears arbitrary and should be further justified. The authors may argue that the pandemic represents a unique context that could attenuate the association between Internet/social media use and mental health issues, and thus should not be included in the analysis. This meta-analysis is relevant: A four-level meta-analytic review of the relationship between social media and well-being: A fresh perspective in the context of COVID-19. (2024) Current Psychology, 43(16), 14972-14986.

8. Given the cross-sectional nature of most studies, I would encourage the authors to refrain the use of causal language. For example, I would suggest the authors to revise the title to: The association between internet addiction and adolescents mental health: A meta-analytic review. Similarly, the abstract should refrain causal language.

9. The comprehensive assessments of heterogeneity, multiple methods of publication bias assessment (funnel plots, trim-and-fill, Egger's test), well-presented forest plots, and the inclusion of moderator analyses are commendable for the rigor. The analysis was nicely done.

Author Response

(The authors gave the same response as above.)

Round 2

Reviewer 2 Report

Comments and Suggestions for Authors

The authors were responsive to all comments.

In my opinion, the paper can be published in the present form.

Best regards

Reviewer 3 Report

Comments and Suggestions for Authors

The authors have addressed all my comments well. I appreciate all their efforts. The paper is ready for publication